# An Innovative Use of Propolis in the Production of Dipping Sauce Powder as a Functional Food to Mitigate Testicular Toxicity Induced by Cadmium Chloride: Technological and Biological Evidence

**DOI:** 10.3390/foods12163069

**Published:** 2023-08-16

**Authors:** Marwa A. Sheir, Francesco Serrapica, Rania A. Ahmed

**Affiliations:** 1Department of Special Food and Nutrition, Food Technology Research Institute, Agricultural Research Center, Giza 3725005, Egypt; marwa.sheir@yahoo.com; 2Department of Agricultural Sciences, University of Naples Federico II, 80055 Portici, Italy; 3Department of Zoology, Faculty of Science, Suez University, Suez 43511, Egypt; rania.sheir@yahoo.com

**Keywords:** propolis, dipping sauce fortification, cadmium chloride toxicity mitigation, rat model, fertility hormones, histology

## Abstract

Propolis is a common natural ingredient used in food production, food packaging, and pharmaceutical products. Therefore, the aim of our study was to prepare dipping sauce powders as an innovative functional product with a regular and spicy taste from economical raw materials with high nutritional value. The developed products were fortified with propolis powder at 250, 500, and 750 mg/kg. All studied dipping sauces were subjected to a palatability test, a nutritional evaluation, and a microbiological assay performed during 6 months of storage. In addition, an in vivo study was designed to evaluate the efficacy of these products in preventing the testicular toxicity disorders induced by cadmium chloride (CdCl_2_) in albino rats. Based on the palatability test, the dipping sauces supplemented with propolis at 250 mg/kg and 500 mg/kg were preferred. Moreover, all samples were safe to consume within 6 months. Biological results showed that all investigated propolis-enriched dipping sauce samples caused an improvement in all CdCl_2_-induced testicular histopathological and biochemical changes, especially the spicy dipping sauce powder fortified with 500 mg/kg of propolis.

## 1. Introduction

Propolis, often known as bee glue, is a waxy natural resinous substance produced by bees (*Apis mellifera*) by combining bee saliva and wax with pollen and plant resins, which honeybees use as a protective coating for inner walls of their hives [1]. Chemically, propolis consists of resins (50%) composed of flavonoids and phenolic acids, waxes (up to 30%), essential oils (10%), pollen (5%), and other organic materials (5%) [1]. Owing to its polar phenolic compounds, especially the flavonoid component, raw propolis and its aqueous and ethanolic extracts have been shown to be antibacterial agents [1]. Due to its antiseptic, anti-inflammatory, antioxidant, antibacterial, antimycotic, antifungal, antidiabetic, antiulcer, anticancer, and immunomodulatory properties, propolis has been used extensively in traditional and alternative medicine since 300 BC for the treatment of a variety of diseases [1]. It has also been used for various purposes, including food production, food packaging, and wound healing [1,2]. However, the use of propolis as a food ingredient is often restricted, mainly due to its low oral bioavailability, in addition to its bitter taste and odor [2]. Humans have been dipping food in sauces for thousands of years [3]. Dipping sauces are eaten all over the world in various forms. Dipping sauces are used by people to add flavor or texture to foods such as crackers, potato chips, falafel, chopped raw seafood, vegetables and fruits, cubed pieces of cheese, and meat and are commonly used with appetizers, finger foods, and other types of food, such as sandwiches [3,4].

Infertility is a significant problem in modern societies, affecting many couples worldwide. There is growing interest in the role of environmental metal exposures in reducing sperm concentration and male fertility in humans [5]. Male factors contribute to infertility in about 40–50% of cases worldwide [6]. The biological effects of heavy toxic metals such as cadmium (Cd) on the human body are well known [5]. Over time, the accumulation of cadmium in various visceral organs such as the liver, lungs, kidneys, and testes has altered many physiological processes and the histological structure of testicular tissue in various animal species [7]. Humans are exposed to Cd through tobacco use, food, and industrial occupational and environmental pollution [7].

For this reason, the aim of this study was to produce a dipping sauce powder fortified with propolis as a functional food to reduce testicular toxicity induced by cadmium chloride (CdCl_2_) using technological, biochemical, histological, and morphometric studies.

## 2. Materials and Methods

### 2.1. Materials

#### 2.1.1. Propolis

Two types of propolis were used: Egyptian propolis (crude) obtained from the Apiary of Beekeeping Research Section, Plant Protection Research Institute, Agriculture Research Center at Dokki, Giza, Egypt, and Chinese propolis powder obtained from Anhui, China. An amount of 50 g of crude Egyptian propolis was extracted with 90% ethanol (500 mL) by mixing for 24 h at room temperature in a dark place. The crude extract was recovered by centrifugation (3000× *g*, 10 min), then filtered to remove particles and obtain a clear extract. The obtained clear extract was concentrated by a rotary vacuum evaporator (Equitron, Roteva, Equitron Medica Private Limited, Mumbai, India) under reduced pressure at 40 °C to eliminate any volatile alcohol. The extract was then freeze-dried in a lyophilizer (Lyovapor L-200, Butchi, Essen, Germany) at −55 °C. The freeze-dried powder sample was stored in laminated pouches.

#### 2.1.2. Plant Materials

Dried tomatoes were obtained from the Egypt-Italian Agency for Agribusiness and Trade Cooperation of Cairo, Egypt. Salt, onion powder, garlic powder, fenugreek powder, ginger powder, curry powder, hot red pepper powder, and yellow split chickpeas were purchased from Metro Market (Nasr City, Egypt).

#### 2.1.3. Shrimp Shell and Head Powder

Shrimp shell and head powder were prepared according to the methods described by Rengga et al. [8]. The shells and heads were washed thoroughly, then dried in a hot air oven at 60 °C for 24 h. These dried samples were finely ground by mixing three times for two minutes/each, then sieved and packed in glass bottles for storage.

#### 2.1.4. Chemicals and Solvents

Cadmium chloride (CdCl_2_) and kits were purchased from Sigma (St. Louis, MO, USA). Ethanol and other chemicals and solvents were purchased from Al-Gomohria Company (Abou-Zabal, Egypt). All other chemicals and solvents were of analytical grade. 

### 2.2. Sauce Preparation and Properties

#### 2.2.1. Preparation of Dipping Sauce Powder Samples and Evaluation of Palatability

The ingredients used to prepare the dipping sauce powders are listed in Table 1. Egyptian propolis was used in three concentrations (250 mg/kg, 500 mg/kg, and 750 mg/kg) to prepare the dipping sauce powders. Two different types of innovative sauces were prepared to suit all tastes: regular sauces (C, C1, C2, and C3) and spicy sauces (Cs, Cs1, Cs2, and Cs3). All the dipping sauce mixes were kept in airtight polyethylene bags prior to technological and biological studies and stored at room temperature (25 ± 5 °C) for six months for the shelf-life assay. 

Ten experienced panelists from the Food Technology Research Institute (Agricultural Research Center, Giza, Egypt) evaluated the palatability of our products according to the procedure described by Amerine et al. [9]. Ten panelists were given eight bowls of dipping sauce powder samples mixed with oil to identify each quality. A hedonic scale of 1 to 9 was used to evaluate the dishes’ taste, color, flavor, and overall acceptability. The dishes were assessed as 9 for excellent, 6 for decent, and below 4 for bad or undesirable.

#### 2.2.2. Chemical and Sensory Properties of Propolis and Nutritional Value of Dipping Sauce Samples

All samples were analyzed for moisture, proteins, fibers, and fats on a dry weight (DW) basis according to the standard procedures recommended by Latimer [10]. Ash content was determined in all dipping sauce samples on a dry weight (DW) basis, as mentioned by Latimer [10]. The percentage of available carbohydrates (on a DW basis) was determined according to Fraser and Holmes [11]. Total flavonoids, total phenolics, and DPPH free radical scavenging activity of Egyptian and Chinese propolis were determined according to Ahn et al. [12], Ghasemzadeh et al. [13], and Elslimani et al. [14], respectively. The color, appearance, and smell of the Egyptian and Chinese propolis were described according to Kosalec et al. [15]. The contents of iron (Fe), phosphorous (P), selenium (Se), and zinc (Zn) were determined using an atomic absorption spectrophotometer (3300 Perkin-Elme) as described by Latimer [10]. The energy content of all dipping sauce samples was calculated using the formula reported by James [16].

#### 2.2.3. Shelf-Life Study of Dipping Sauce Powder Samples

Dipping sauce powder samples were stored at room temperature (25 ± 5 °C) throughout the study period of 6 months, and their microbial quality was evaluated. Total bacterial and pathogenic counts of coliforms and *Salmonella* were performed during different storage periods according to the procedures described by Harwitz [17] and Andrews et al. [18], respectively.

### 2.3. Biological Study Design

We are investigating the effect of our dipping sauce powders as a functional food to ameliorate testicular toxicity induced by cadmium chloride. The study was conducted according to the ethical guidelines for the use and care of laboratory animals and approved by the local ethics committee of Suez University (approval number 22319). 

A total of 48 healthy adult male Sprague–Dawley rats aged approximately 3 months and weighing between 180 and 220 g were used. The rats were obtained from the National Research Centre of Giza (Egypt) and were maintained under ideal sanitary conditions. Feed and water were provided ad libitum. The basal diet consisted of 15% casein, 10% corn oil, 4% salt mix, 1% vitamin mix, 5% cellulose, and 65% starch [19]. All rats were acclimatized to the experimental conditions for one week. This was also done to exclude any infection. After the acclimation period, the rats were randomly divided into eight equal groups (6 rats/group) as follows: Group I (negative control): Rats were fed the basal diet for 30 days;Group II (positive control): Rats were fed the basal diet and received CdCl_2_ solution orally via stomach tube every other day for 30 days at a dose of 5 mg/kg body weight [20];Group III: Rats were fed the basal diet containing 5% C formula and received CdCl_2_ solution orally via stomach tube every other day for 30 days at a dose of 5 mg/kg body weight;Group IV: Rats were fed the basal diet containing 5% C1 formula and received CdCl_2_ solution orally via stomach tube every other day for 30 days at a dose of 5 mg/kg body weight;Group V: Rats were fed the basal diet containing 5% C2 formula and received CdCl_2_ solution orally via stomach tube every other day for 30 days at a dose of 5 mg/kg body weight;Group VI: Rats were fed the basal diet containing 5% Cs formula and received CdCl_2_ solution orally via stomach every other day for 30 days at a dose of 5 mg/kg body weight;Group VII: Rats were fed the basal diet containing 5% Cs1 formula and received CdCl_2_ solution orally via stomach tube every other day for 30 days at a dose of 5 mg/kg body weight;Group VIII: Rats were fed the basal diet containing 5% (Cs2) formula and received CdCl_2_ solution orally via stomach tube every other day for 30 days at a dose of 5 mg/kg body weight.

#### 2.3.1. Biochemical Analysis

After 30 days, the rats were fasted overnight, then sacrificed, and blood was collected immediately from the portal vein in clean and dried tubes. Blood samples were centrifuged at 3000 rpm for 15 min to obtain serum; then, sera were stored at 20 °C until assayed by enzyme-linked immunosorbent assay (ELISA) for the testosterone, luteinizing hormone (LH), and progesterone concentrations [21]. For the assessment of oxidative stress in testicular tissues, clear supernatant was separated and used for malondialdehyde (MDA) [22] and total antioxidant capacity (TAC) [23] analysis.

#### 2.3.2. Histopathological and Histomorphometric Examinations 

All animals were dissected at the end of the experimental period, and their right testes were removed, fixed in 10% neutral formalin, dehydrated, cleared, and embedded in paraffin wax. Paraffin sections of 5 µm thickness were prepared and stained with routine hematoxylin and eosin stain [24]. For morphometric evaluation, the diameter and germinal epithelial height of the seminiferous tubules were measured. Epithelial height was measured from the spermatogenic cells on the inner surface of the basement membrane through the most advanced cell types lining the lumen of the tubules. For analysis, five fields per testis section were taken at ×100 (10 objective ×10 ocular).

### 2.4. Statistical Analysis 

Statistical analysis was performed using SPSS One-Way ANOVA, version 25 (IBM Corp., Armonk, NY, USA, 2013). The data were treated as a complete randomized design [25]. Multiple comparisons were performed using the Duncan test. The significance level was *p* < 0.05.

## 3. Results and Discussion

### 3.1. Technological Results

#### 3.1.1. Functional and Technological Properties of Egyptian Propolis

The chemical composition, antioxidant properties, and some sensory characteristics of the Egyptian and Chinese propolis powders are summarized in Table 2. Compared to the Chinese propolis powder, the Egyptian propolis powder had significantly (*p* > 0.05) higher levels of protein, fiber, and antioxidants (total phenolics, total flavonoids, and DPPH), in addition to having a non-aromatic smell. These results are consistent with those reported in other studies [15,26], which found that Egyptian and Chinese propolis have differences in odor and color. This could be due to the geographical differences between Chinese and Egyptian propolis, since the origin of propolis affects the color of the substance. In addition, the Egyptian propolis was previously reported as significantly higher in fiber, protein, flavonoids, total phenolics, and DPPH contents than Chinese propolis [26]. These results confirm the health benefits of Egyptian propolis compared to Chinese propolis, and its addition could be a better choice in food processing.

#### 3.1.2. Palatability Tests, Nutritive Values, and Shelf-Life Assay of Investigated Products

To select the best formula and ratio of raw materials for dipping sauce mixes, a palatability experiment was designed. According to the preference results (Figure 1 and Figure 2), the regular and spicy control samples (C and Cs, respectively) and the dipping sauce powders fortified with 250 mg/kg propolis (C1 and Cs1) showed higher scores in color, flavor, taste, and overall acceptability.

Moreover, it was observed that the acceptance in the regular and spicy sauce samples decreased with increased concentration of propolis (C2 and Cs2) for all parameters. The sample with a concentration of 750 mg/kg of propolis was rated as unacceptable. Since samples C, C1, C2, Cs, Cs1, and Cs2 were within the acceptable range, we selected them for our studies. These results are in line with the study of Segueni et al. [27], who reported that the unpleasant taste and smell of propolis may limits its use by affecting the sensory evaluation aspects of food products, such as color, odor, appearance, texture, and overall acceptability.

Nutritional interventions may be essential to maintain male fertility and testosterone levels, which are directly related to sperm mitochondrial activity, a significant determinant of sperm quality. In addition, this effect depends on dietary factors that are both quantitative and qualitative, such as the number of calories in each macronutrient (carbohydrates, proteins, and fats), as well as in micronutrients, such as zinc, selenium, iron, and phosphorus [28,29].

Table 3 and Table 4 summarize the macronutrients and micronutrients of the dipping sauce mixes. The results show that supplementation of dipping sauce mixes with Egyptian propolis powder increased the values of nutrient and mineral intake compared to the control samples (C and Cs), but the increase was significant only in the fibers and phosphorous contents for all samples; in the protein, ash, and selenium contents only in regular samples (C1 and C2); and in zinc in regular sample (C2). Moreover, significant increases in ash, zinc, and selenium contents were also observed in the spicy sauce sample (Cs2). This makes it a good option to include in our diets, especially for those suffering from micronutrient deficiencies. As a result, Egyptian propolis can be used as a nutritious ingredient in the production of healthy foods.

The shelf life of dipping sauce samples depends on a number of factors, such as expiry date, method of preparation, and storage conditions. In our study, different microorganisms (i.e., total bacterial count, *Salmonella* spp., and coliform bacteria) were identified in the dipping sauce mixes partially supplemented with Egyptian propolis over a storage period of 6 months. It is worth mentioning that pathogens such as *Salmonella* spp. and coliform were not detected in all samples during the storage period.

The aerobic plate count method was used to assess the rate of bacterial growth in dipping sauce samples during storage. The results (Figure 3) show that all dipping sauce samples complied with the NSW Food Authority requirements. The unacceptable limit of aerobic plate count is ≥5 log cfu/mL [30]. Our investigations showed that propolis fortification achieved an average reduction in total bacterial count compared to the control samples (C and Cs), which may be due to its high content of polyphenolic compounds. Propolis has many uses in different fields, including therapeutic applications, as well as active ingredients and food packaging. Additionally, propolis has been found to improve the shelf life of several foods, including vegetables, fruits, and beverages, and can be used as a safe, natural preservative in meat, fish, and poultry [27].

### 3.2. Biological Results

#### 3.2.1. Histological and Morphometric Results 

As depicted in Figure 4, examination of the testes of control rats revealed a typical testicular architecture with normal seminiferous tubules lined by an orderly sequence of spermatogenic cells (spermatogonia, spermatocytes, spermatids, and sperm), Sertoli cells, and intertubular connective tissue. The histoarchitecture of testicular tissue was significantly altered by CdCl_2_ treatment; the basal lamina showed marked thickness, some seminiferous tubules shrank significantly, and the remaining tubules were atrophied and devoid of germ cells. Separation of seminiferous tubules and degradation of interstitial tissue were observed. When the testes of animals treated with CdCl_2_ were compared to the controls, image analysis data showed a significant decrease in seminiferous tubule diameter and epithelial height. All cotreated groups showed varying degrees of improvement when compared to testis sections from animals exposed to CdCl_2_ alone, with significant increases in the seminiferous tubule diameter and epithelial height. When compared to all other cotreated groups, animals in group III showed a slight improvement in testicular tissue, while group VI showed a moderate improvement in testicular tissue. On the other hand, the testicular tissue in groups IV, V, VII, and VIII showed a marked improvement, particularly in group VIII, which had testicular tissues that were within the normal structure, with normal interstitial cells, testicular tubule configurations, cellularity, and sperm density and orientation (Figure 4 and Table 5).

#### 3.2.2. Biochemical Results

According to the data shown in Table 6, CdCl_2_ treatment resulted in significant decreases in testosterone, LH, and progesterone levels, as well as TAC levels, while MDA values increased significantly (*p* > 0.05). Comparing the cotreated groups, the levels of testosterone, LH, progesterone, and TAC increased significantly (*p* < 0.05) from group III to group VIII. In contrast, an opposite trend was observed for MDA concentration, whose levels decreased significantly (*p* < 0.05) across the cotreated groups, reaching the lowest concentration in group VIII.

Recent histopathological findings are consistent with those of El-Habibi et al. [31], El-Neweshy et al. [32], and Eleawa et al. [33], who reported that CdCl_2_ caused testicular lesions, seminiferous tubule damage, necrosis, Leydig and Sertoli cell degeneration, and a reduction in the spermatozoa population. Furthermore, CdCl_2_ induced a disorganized germinal epithelium with significant vacuolation, thickened interstitial cells with oedema, congested blood vessels, and fragmented germ cells with pyknotic nuclei. Through a breach in the blood–testis barrier, Cd enters the seminiferous tubules and causes localized testicular necrosis and dystrophy, resulting in a reduction in the number of germ cells and infertility [34]. 

El-Habibi et al. [31] and Yang et al. [35] both confirmed that Cd causes testicular oxidative stress by inducing a large decrease in the levels of sex hormones (T, E2, FSH, and LH) in rats, in addition to a large increase in testicular MDA levels in conjunction with a significant decrease in TAC [33]. The authors found that increased oxidative stress and the apoptotic pathway cause decreased spermatogenesis. As suggested by Qadori and Al-shaikh [36] and Dimer et al. [37], reduced sperm quality could possibly be caused by ROS-induced membrane damage, macromolecular degradation, or a dysfunctional H_2_O_2_ scavenging mechanism inhibiting steroidogenesis in Leydig cells due to H_2_O_2_ accumulation. Ikediobi et al. [38] provided evidence of the deleterious effects of CdCl_2_ on the testes, including the known depletion of glutathione and protein-bound sulfhydryl groups, leading to increased generation of reactive oxygen species (ROS), including superoxide ions, hydroxyl radicals, and hydrogen peroxide. El-Neweshy et al. [32] observed a significant increase in testicular oxidative stress, as evidenced by a significant increase in MDA and a significant decrease in GSH, which resulted in irreparable damage to the testicular cells. Cadmium has been shown to reduce testicular biochemical function and the steroidogenic activities of the testes, in addition to causing oxidative stress, germ cell death, and cadmium-induced autophagy, which is directly linked to a decline in male reproductive capacity [39]. 

According to Abu-Almaaty et al. [40] and El-Amawy et al. [41], propolis has a protective effect on different organs of albino rats due to its antioxidant function against oxidative stress caused by heavy metals; moreover, propolis improved the histopathology and the morphometry of the testicular tissues in groups treated with propolis-containing diets. Hashem [42] studied the preventive properties of propolis against testicular damage caused by carbon tetrachloride (CCl_4_) and found that propolis increased testosterone levels while decreasing oxidant levels. In addition, propolis improved the normal histological structure of the seminiferous tubules, minimally congested testicular blood vessels, and significantly activated spermatogenesis in a group treated with propolis, along with CCl_4_.

In rabbits treated with triphenyltin, Yousef et al. [43] found that propolis had antioxidant properties. This finding was supported by histological evidence showing the typical histoarchitecture of the testes in the propolis-treated group. Similar conclusions were reached by Abd-Elrazek et al. [44], who found that coadministration of propolis reduced the effects of tramadol on testicular oxidant balance, induced a significant decrease in MDA while increasing SOD and GSH levels, and reduced the effects on tramadol on blood oxidant balance. By reducing elevated levels of caspases, particularly caspase-3, propolis therapy was found to reduce cadmium-induced apoptotic cells in the testes in situations of induced reproductive toxicity [45,46,47]. It also increased sperm count and gonadotropin levels compared to the tramadol-treated group. Nitric oxide generation in the testicular tissue of male rats has been shown to increase in response to propolis [48], which may improve spermatogenesis. The ability of propolis to prevent testosterone loss caused by exogenous toxins has been investigated in several studies [42,45,49]. One of the bioactive components of propolis, the flavonoid chrysin, has been shown to increase testosterone production in the testes [50]. At the molecular level, by increasing the mRNA and protein levels of the steroidogenic acute regulatory protein (StAR), propolis was found to increase steroidogenesis in the testes of diabetic rats, cytochrome P450 A1 (CYP11A1; an enzyme that turns cholesterol into pregnenolone), cytochrome P450 17A1 (CYP17A1), 3β-hydroxysteroid dehydrogenase (3β-HSD), and 17β-hydroxy-steroid dehydrogenase (17β-HSD) [51].

## 4. Conclusions

In conclusion, our results show that Egyptian propolis possesses antioxidant and antimicrobial properties and has a non-aromatic smell compared to Chinese propolis powder. Moreover, it can be successfully used as a food additive to produce dipping sauce powders (regular or spicy) at levels of 250 mg/kg and 500 mg/kg to improve nutritional value and technological properties. Furthermore, propolis ameliorated all testicular histopathological and biochemical alterations induced by CdCl_2_, especially the spicy dipping sauce powder fortified with 500 mg/kg (CS2).

## Figures and Tables

**Figure 1 foods-12-03069-f001:**
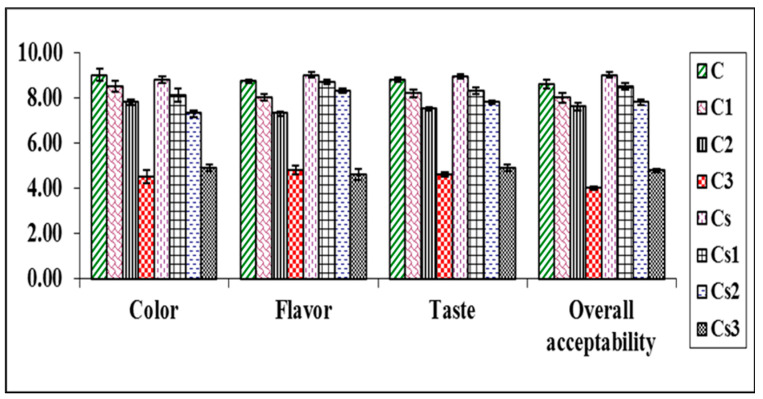
Acceptability scores of dipping sauce formulations.

**Figure 2 foods-12-03069-f002:**
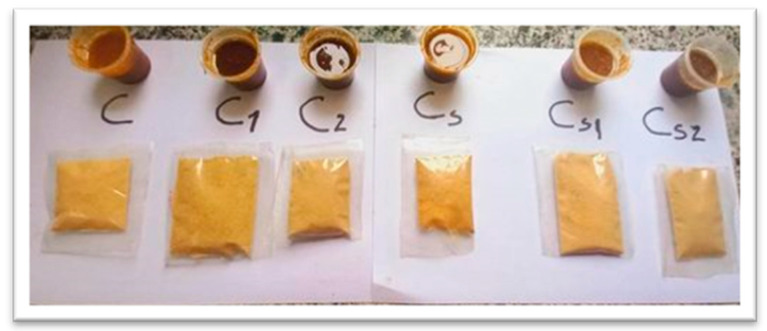
Acceptable dipping sauce formulations (regular samples (C–C2) and spicy samples (Cs–Cs2)).

**Figure 3 foods-12-03069-f003:**
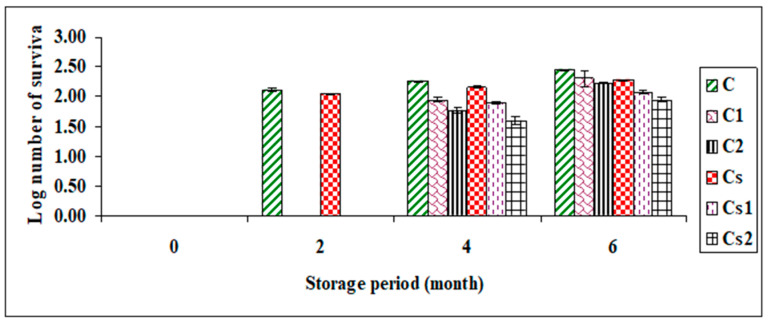
Total bacterial count at 25 ± 5 °C expressed as log (CFU/mL) for up to 6 months of storage.

**Figure 4 foods-12-03069-f004:**
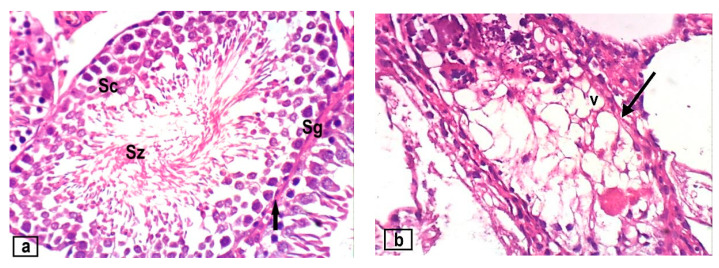
(**a**) Section of testis of a control rat showing normal seminiferous tubules (arrow), spermatogonia (Sg), spermatocytes (Sc), and spermatozoa (Sz). (**b**) Section in testis of a rat treated with CdCl2 showing an atrophoid seminiferous tubule with marked degeneration and necrosis of spermatogonial cells and vacuolation (v) and thickened tubular wall (arrow). (**c**) Section in testis of a rat from group III, showing mild restoration of spermatogonial cells (arrow), spermatids (St). (**d**) Section in testis of a rat from group IV, showing marked improvement of testicular tissue with dome degeneration (star). (**e**) Section in testis of a rat from group V, showing marked improvement of testicular tissue. (**f**) Section in testis of a rat from group VI, showing moderate improvement of testicular tissue with normal spermatogonia (arrow), with absence of spermatozoa (star). (**g**) Section in testis of a rat from group VII, showing marked improvement of testicular tissue. (**h**) Section in testis of a rat from group VIII, showing normal testicular tissue with complete spermatogenesis. (H&E stain; ×400).

**Table 1 foods-12-03069-t001:** Dipping sauce powder formulations.

	Formulations	C	C1	C2	C3	Cs	Cs1	Cs2	Cs3
Ingredients									
Propolis powder (mg/kg)		-	293.2	586.5	879.7	-	304.8	609	913.5
Yellow split chickpeas (g)		250	250	250	250	250	250	250	250
Fenugreek powder (g)		15	15	15	15	15	15	15	15
Shrimp shell powder (g)		400	400	400	400	400	400	400	400
Red hot pepper (g)		-	-	-	-	45	45	45	45
Garlic powder (g)		10	10	10	10	10	10	10	10
Onion powder (g)		5	5	5	5	5	5	5	5
Curry powder (g)		18	18	18	18	18	18	18	18
Salt (g)		10	10	10	10	10	10	10	10
Ginger (g)		15	15	15	15	15	15	15	15
Dried tomato powder (g)		450	450	450	450	450	450	450	450

C = regular dipping sauce without propolis; C_1_ = regular dipping sauce with 250 mg/kg propolis; C_2_ = regular dipping sauce with 500 mg/kg propolis; C_3_ = regular dipping sauce with 750 mg/kg propolis; Cs = spicy dipping sauce without propolis; Cs1 = spicy dipping sauce with 250 mg/kg propolis; Cs2 = spicy dipping sauce with 500 mg/kg propolis; Cs3 = spicy dipping sauce with 750 mg/kg propolis.

**Table 2 foods-12-03069-t002:** Chemical composition and antioxidant and sensory properties of Egyptian and Chinese propolis (mean ± SE).

Chemical Composition	Egyptian Propolis	Chinese Propolis
Moisture (%)	4.11 ± 0.28 ^a^	5.68 ± 0.69 ^a^
Proteins (%)	10.25 ± 0.16 ^a^	8.98 ± 0.29 ^b^
Fats (%)	21.54 ± 0.49 ^b^	41.68 ± 0.90 ^a^
Fibers (%)	49.87 ± 0.18 ^a^	20.48 ± 0.42 ^b^
Available carbohydrates (%)	14.23 ± 0.26 ^b^	23.18 ± 1.22 ^a^
**Antioxidant Properties**		
Total phenolic content (mg GAE/g sample DW)	200.70 ± 1.54 ^a^	135.26 ± 5.92 ^b^
Total flavonoid content (mg quercetin/g sample DW)	91.86 ± 2.51 ^a^	29.03 ± 3.25 ^b^
DPPH scavenging activity (%)	78.77 ± 0.82 ^a^	66.70 ± 0.78 ^b^
**Sensory Properties**		
Color	Brown	Tortilla brown
Smell	Not aromatic	Aromatic
Appearance	Dry	Dry

^a^, ^b^ The same superscript letter in a row indicates no significant difference (*p* > 0.05) between means.

**Table 3 foods-12-03069-t003:** Macronutrient contents (mean ± SE) of dipping sauce mixes.

Sample	Protein	Fat	Ash	Fibers	Available Carbohydrate	Energy
	%	%	%	%	%	Kcal
C	21.20 ± 0.06 ^c^	7.97 ± 0.27 ^a^	4.12 ± 0.02 ^e^	2.51 ± 0.01 ^f^	64.21 ± 0.32 ^a^	413.22 ± 2.13 ^a^
C1	21.45 ± 0.05 ^b^	8.02 ± 0.27 ^a^	4.29 ± 0.02 ^d^	2.63 ± 0.01 ^d^	63.62 ± 0.32 ^ab^	412.42 ± 1.26 ^a^
C2	21.48 ± 0.05 ^b^	8.07 ± 0.27 ^a^	4.47 ± 0.04 ^c^	2.76 ± 0.01 ^b^	63.25 ± 0.25 ^b^	411.43 ± 1.35 ^a^
Cs	22.34 ± 0.02 ^a^	8.47 ± 0.27 ^a^	5.05 ± 0.01 ^b^	2.58 ± 0.01 ^e^	61.54 ± 0.32 ^c^	411.81 ± 1.43 ^a^
Cs1	22.37 ± 0.09 ^a^	8.59 ± 0.23 ^a^	5.11 ± 0.01 ^b^	2.71 ± 0.01 ^c^	61.26 ± 0.27 ^c^	411.65 ± 1.10 ^a^
Cs2	22.47 ± 0.02 ^a^	8.67 ± 0.24 ^a^	5.23 ± 0.02 ^a^	2.83 ± 0.01 ^a^	60.80 ± 0.26 ^c^	411.12 ± 1.17 ^a^
LSD at 0.05	0.17	0.80	0.07	0.03	0.90	4.46

^a^, ^b^, ^c^, ^d^, ^e^, and ^f^ The same superscript letter in a column indicates no significant difference (*p* > 0.05) between means.

**Table 4 foods-12-03069-t004:** Micronutrient contents (mean ± SE) of dipping sauce mixes.

Sample	Micronutrient Contents			
	P	Fe	Zn	Se
C	57.83 ± 0.17 ^e^	111.67 ± 0.88 ^b^	146.05 ± 0.13 ^e^	1.33 ± 0.01 ^d^
C1	59.50 ± 0.29 ^cd^	111.83 ± 0.86 ^b^	147.05 ± 0.64 ^e^	1.36 ± 0.00 ^c^
C2	60.50 ± 0.29 ^bc^	113.00 ± 0.15 ^ab^	148.71 ± 0.38 ^d^	1.39 ± 0.00 ^b^
Cs	58.50 ± 0.76 ^de^	112.65 ± 0.31 ^ab^	149.71 ± 0.57 ^cd^	1.39 ± 0.01 ^b^
Cs1	61.27 ± 0.03 ^ab^	113.23 ± 0.09 ^ab^	151.05 ± 0.13 ^bc^	1.41 ± 0.01 ^ab^
Cs2	61.80 ± 0.06 ^a^	113.87 ± 0.03 ^a^	153.05 ± 0.87 ^a^	1.43 ± 0.01 ^a^
LSD at 0.05	1.11	1.61	1.63	0.02

^a^, ^b^, ^c^, ^d^, and ^e^ The same superscript letter in a column indicates no significant difference (*p* > 0.05) between means.

**Table 5 foods-12-03069-t005:** Mean of seminiferous tubule diameter and epithelial height (μm) in the different experimental groups (mean ± SE).

Group	Seminiferous Tubule Diameter	Epithelial Height
GI (Control)	394.00 ± 2.08 ^a^	124.00 ± 0.58 ^a^
GII (CdCl_2_)	254.48 ± 1.16 ^g^	55.70 ± 0.93 ^g^
GIII	322.85 ± 1.25 ^f^	86.31 ± 0.66 ^f^
GIV	366.74 ± 1.85 ^d^	113.67 ± 0.88 ^d^
GV	370.08 ± 0.56 ^d^	116.00 ± 0.58 ^c^
GVI	333.77 ± 2.04 ^e^	102.00 ± 1.15 ^e^
GVII	376.21 ± 1.17 ^c^	118.00 ± 0.58 ^bc^
GVIII	388.67 ± 1.20 ^b^	119.67 ± 0.88 ^b^
LSD at 0.05	5.50	2.96

^a^, ^b^, ^c^, ^d^, ^e^, ^f^, and ^g^ The same superscript letter in a column indicates no significant difference (*p* > 0.05) between means.

**Table 6 foods-12-03069-t006:** Changes in serum testosterone, LH and progesterone hormones, and testicular MDA and TAC in the investigated groups (mean ± SE).

Group	Testosterone	LH	Progesterone	MDA	TAC
	(ng/mL)	(mIU/mL)	(ng/mL)	(nmol/g tissue)	(nmol/g tissue)
GI (Control)	2.89 ± 0.01 ^a^	2.92 ± 0.02 ^a^	1.86 ± 0.01 ^a^	39.44 ± 0.70 ^g^	36.18 ± 0.57 ^a^
GII (CdCl_2_)	1.38 ± 0.02 ^g^	2.02 ± 0.03 ^g^	1.03 ± 0.03 ^e^	99.45 ± 0.53 ^a^	12.83 ± 0.20 ^e^
GIII	1.87 ± 0.03 ^f^	2.19 ± 0.01 ^f^	1.28 ± 0.01 ^d^	72.62 ± 0.26 ^b^	13.62 ± 0.26 ^d^
GIV	2.26 ± 0.02 ^d^	2.34 ± 0.04 ^e^	1.66 ± 0.04 ^b^	56.76 ± 1.49 ^d^	18.57 ± 0.35 ^d^
GV	2.31 ± 0.02 ^c^	2.51 ± 0.01 ^d^	1.63 ± 0.01 ^b^	53.21 ± 0.48 ^e^	18.63 ± 0.20 ^d^
GVI	2.00 ± 0.03 ^e^	2.34 ± 0.04 ^e^	1.47 ± 0.02 ^c^	59.54 ± 0.64 ^c^	18.65 ± 0.32 ^d^
GVII	2.36 ± 0.02 ^c^	2.62 ± 0.01 ^c^	1.65 ± 0.03 ^b^	54.18 ± 1.52 ^de^	22.77 ± 0.38 ^c^
GVIII	2.54 ± 0.03 ^b^	2.77 ± 0.02 ^b^	1.79 ± 0.01 ^a^	43.54 ± 0.89 ^f^	25.25 ± 0.64 ^b^
LSD at 0.05	0.07	0.08	0.07	2.77	1.19

^a^, ^b^, ^c^, ^d^, ^e^, ^f^, and ^g^ The same superscript letter in a column indicates no significant difference (*p* > 0.05) between means.

## Data Availability

The corresponding author can provide the data used in this study upon request.

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
