# Peer review of "An Innovative Use of Propolis in the Production of Dipping Sauce Powder as a Functional Food to Mitigate Testicular Toxicity Induced by Cadmium Chloride: Technological and Biological Evidence"

_foods, 2023, doi:10.3390/foods12163069_

Round 1

Reviewer 1 Report

This manuscript describes the use of two types of propolis to prepare a powder (two types) for sauce, evaluating the sensory characteristics, antioxidants, shelf life, effect on cadmium damage in three different concentrations supplied with the diet and subsequent evaluation of said effect on histology and biochemical parameters. The topic is interesting and the application of this powder to prevent cadmium damage to the testicles is novel. However, the subject is so broad that, in my opinion, a description and discussion of the results is lacking. I mean, every topic is covered very superficially. Authors should consider expanding the results and discussion section.

There are some specific observations:

Title should mention cadmium, or something related to this application.

Line 69 it should say “described”

Line 298 instead of [37] it should be used Author [37]

Tables 5 and 6 should use similar nomenclature as the method section while describing the groups. It should be used G1, G2…. Or GI, GII, etc, but not a mixture.

Finally, I suggest that an English language check be done on the entire manuscript.

English is fine, there are only minor changes suggested.

Author Response

We thank the reviewers for the time spent in reviewing our manuscript. Please note that your specific requests are highlighted in yellow. Other changes are in track change mode. Moreover, a graphic abstract has also been added.

  • Title should mention cadmium, or something related to this application.

AU Thanks for the suggestion. We addressed the reviewer’s comment as follows: “An Innovative Use of Propolis in the Production of Dipping Sauces Powder as a Functional Food to Mitigate Testicular Toxicity Induced by Cadmium: Technological and Biological Evidence”.

  • Line 69 it should say “described” (L 76 in the revised version).

AU Thanks for the remarks. We have changed.

  • Line 298 instead of [37] it should be used Author [37].

AU Thanks for the remarks. According to the reviewer suggestion, we have changed as follows: “As suggested by Qadori and Al-shaikh, [36] and Dimer et al. [37], the reduced sperm quality could possibly be caused by ROS-induced membrane damage or macromolecular degradation or a dysfunctional H2O2 scavenging mechanism inhibiting steroidogenesis in Leydig cells due to H2O2 accumulation” (L 325 in the revised version).

  • Tables 5 and 6 should use similar nomenclature as the method section while describing the groups. It should be used G1, G2…. Or GI, GII, etc, but not a mixture.

AU Thanks for the remarks. We have changed as suggested.

Reviewer 2 Report

In this paper, propolis was incorporated into the dipping sauces, and an evaluation of its physicochemical properties and potential for reducing testicular toxicity in the resulting product was conducted. This study presents intriguing findings; however, there are certain issues that must be addressed.

 1.      Could you please explain the reason for selecting dipping sauces powder as a medium to incorporate propolis? Is there any correlation between dipping sauces powder and fertility disorders? Please provide some explanation in the Introduction.

2.      The final paragraph of the Introdution should also include a brief overview about your research content.

3.      Section 2.1.4 should contain all the chemicals or reagents involved in your research.

4.      Table 1, a footnote is necessary to explain C, C1, C2, C3, Cs, Cs1, Cs2 and Cs3.

5.      Figure 1, some of the bar charts are missing error bars.

6.      Line 87~90, what are the criteria for scoring these products?

7.      Figure 2, it can be observed that there is little color variation among the solid powders, but significant color variation is observed after dissolution in liquids. What could be the reason for this?

8.      Figure 3, the indication of the ordinate is not clear, and the error bar is missing. Have duplicate samples been established?

9.      “Fertility” has been mentioned in multiple instances throughout the text. However, fertility encompasses a comprehensive conceptual framework. It is important to note that this study solely focuses on evaluating testicular toxicity and does not provide sufficient evidence to support the product's potential in enhancing fertility. Therefore, some modifications are suggested for the text.

Moderate editing of English language required

Author Response

We thank the reviewer for the positive feedback, and the competent work which has helped us to clarify our manuscript. Please note that your specific requests are highlighted in green. Other changes are in track change mode. Moreover, a graphic abstract has also been added.

  • Could you please explain the reason for selecting dipping sauces powder as a medium to incorporate propolis? Is there any correlation between dipping sauces powder and fertility disorders? Please provide some explanation in the Introduction.

AU The dipping powder was chosen as an instant product and it has low moisture content, so it can be stored for a long time. It is also suitable in its manufacture with the properties of the materials used in production, and the dipping sauce is one of the side dishes that are served alongside many daily dishes and has great acceptance. According to reviewer suggestion, we have provided some explanation in the Introduction (L 42-47), as follows: “Humans have been dipping foods into sauces for millennia [3]. Dip sauces in various forms are eaten all over the world and people have been using sauces for dipping to add flavor or texture to food such as crackers, potato chips, falafel, chopped raw seafood, vegetables and fruits, cubed pieces of cheese, meat and commonly used for appetizers, finger foods and other food types as sandwiches [3,4].”

  • The final paragraph of the Introduction should also include a brief overview about your research content.

AU Thank you for the useful suggestion. According to reviewer suggestion, we implemented the Introduction as follows (L 59): “For this reason, the aim of this study was to produce a dipping sauce powder fortified with propolis as a functional food to reduce testicular toxicity induced by cadmium chloride (CdCl2) using technological, biochemical, histological, and morphometric studies”.

  • Section 2.1.4 should contain all the chemicals or reagents involved in your research.

AU Thank for this remark. The sentence as rearranged as follows: “Cadmium chloride (CdCl2) and kits were purchased from Sigma (St. Louis, MO, USA). Ethanol and other chemicals and solvents were purchased from Al-Gomohria Company (Abou-Zabal, Egypt). All other chemicals and solvents were of analytical grade” (L 81-84 in the revised version).

  • Table 1, a footnote is necessary to explain C, C1, C2, C3, Cs, Cs1, Cs2 and Cs3.

AU Thank for this remark. Table footnotes has been addedd (L 102-104): C= Regular dipping sauce without propolis; C1= Regular dipping sauce with 250 mg /kg propolis; C2= Regular dipping sauce with 500 mg /kg propolis; C3= Regular dipping sauce with750 mg /kg propolis ;CS= spicy dipping sauce without propolis; CS1= spicy dipping sauce with 250 mg /kg propolis; CS2= spicy dipping sauce with 500 mg /kg propolis; CS3= spicy dipping sauce with 750 mg /kg propolis”.

  • Figure 1, some of the bar charts are missing error bars.

AU Thank for this remark. As suggested, the missing error bars has been addedd.

  • Line 87~90, what are the criteria for scoring these products?

AU The sensory characteristics were asessed according to the standard procedure suggested by Amerine et al., (1965) [Amerine M.A., Pangborn R.M. and Rocssler E. (1965). Principles of sensory evolution of foods. 349, Academic press, New York]; a Hedonic scale of 1 to 9 was used to evaluate the dishes' taste, color, flavor, and overall acceptability. The dishes were assessed as 9 for excellent, 6 for decent, and below 4 for bad or undesirable range.

  • Figure 2, it can be observed that there is little color variation among the solid powders, but significant color variation is observed after dissolution in liquids. What could be the reason for this?

AU Thank for this remark. This is due to the addition of oil, which, in turn, promote an emulsion form and, thus, caused the appearance of the sensory properties of the product such as color, taste, and flavor.

  • Figure 3, the indication of the ordinate is not clear, and the error bar is missing. Have duplicate samples been established?

AU Thank for this remark. According to the reviewer's suggestions, we have rearranged the figure by providing both error bars and ordinate information. Moreover, duplicate samples have been established.

Rev. “Fertility” has been mentioned in multiple instances throughout the text. However, fertility encompasses a comprehensive conceptual framework. It is important to note that this study solely focuses on evaluating testicular toxicity and does not provide sufficient evidence to support the product's potential in enhancing fertility. Therefore, some modifications are suggested for the text.

AU Thank you for your suggestion. As proposed, we changed the term “fertility” (too generic) to “testicular toxicity” throughout the text (e.g., L 17, 57, 127, etc.).

Reviewer 3 Report

·         In the study, two different propolis were compared. It is not clear whether these propolis samples were chosen randomly, and on what basis. Why is it compared to Chinese propolis? Was this Chinese propolis commercially purchased? Egyptian propolis reflects my whole country, only one region?

·         Protein, fat, fiber and sugar were measured in propolis. By what methods were they determined? Is it true that there is such a high protein and sugar in propolis?

·         It is difficult to understand the formulation and groups in Table 1. It should be more descriptive.

·         How were the 250 mg /kg, 500 mg /kg, and 750 mg /kg of Propolis concentrations selected in Table 1? In raw propolis/kg? It is not clear whether this is as a solvent or not.

·         are the fat, protein, etc. ratios in the data in Table 2 given as %? If given in %, isn't it too high?

·         It should be more accurate to give the total phenolic substance amounts in ethanolic extracts obtained from propolis samples as mg GAE/mL in Table 2.

·         In table 2, color values are not measured visually, but with a device, such as Hunter Lab.

·         The purpose of the study is not fully understood. Extending the shelf life of these commercial sauces or increasing their biological activity? Why infertile was chosen as the biological activity test in the study?

The article needs a simpler explanation

Author Response

We sincerely thank the reviewer for his competent work and valuable comments, which have helped to clarify our manuscript. Please note that your specific requests are highlighted in blue. Other changes are in track change mode. Moreover, a graphic abstract has also been added.

  • In the study, two different propolis were compared. It is not clear whether these propolis samples were chosen randomly, and on what basis. Why is it compared to Chinese propolis? Was this Chinese propolis commercially purchased? Egyptian propolis reflects my whole country, only one region?

AU Thanks for the remarks. The Egyptian kind was chosen because it is the local type and is produced locally, while the Chinese kind was chosen because it is the imported type, the most widely circulated, and the cheapest price.

  • Protein, fat, fiber, and sugar were measured in propolis. By what methods were they determined? Is it true that there is such a high protein and sugar in propolis?

AU The sugars in propolis have not been determined, while the methods used to assess each component are highlighted in the materials and methods section, about protein content. The obtained results are reported to the dry weight, and there are many studies that agree with the values obtained in our study, such as for example, Abd El-Hady et al. (2020) about the Egyptian and Chinese propolis.

  • It is difficult to understand the formulation and groups in Table 1. It should be more descriptive.

AU Thanks for the remarks. The reviewer’s suggestions have been explained by adding footnotes to the table.

  • How were the 250 mg /kg, 500 mg /kg, and 750 mg /kg of Propolis concentrations selected in Table 1? In raw propolis/kg? It is not clear whether this is as a solvent or not.

AU Thanks for the remark. Propolis, as well as the other ingredients, have been used as dry raw powder and, as such, were dosed as g/kg of powder of propolis in preparing the different formulations.

  • … are the fat, protein, etc. ratios in the data in Table 2 given as %? If given in %, isn't it too high?

AU Thanks for the remarks. The results listed in Table 2 are expressed as % of dry weight and, as far we as know, are close to literature values obtained for similar products (e.g., Abd El-Hady et al. (2020; EFFECT OF THE EGYPTIAN PROPOLIS ON THE BIOACTIVE COMPOUNDS CONTENT IN TOMATO PLANTS. Zagazig Journal of Agricultural Research, 47(2), 579-586.)

  • It should be more accurate to give the total phenolic substance amounts in ethanolic extracts obtained from propolis samples as mg GAE/mL in Table 2.

AU Thanks for the remarks. We have expressed total phenolic content as mg on a dry weight basis since we have used for analyses a dried extract rather than a liquid extract. Indeed, the Chinese Propolis Powder was obtained from Anhui, China. 50 g of crude Egyptian propolis was extracted with 90% ethanol (500 ml) by mixing for 24 h at room temperature in a dark place. The crude extract was recovered by centrifugation (3000 g, 10 min) and dried under vacuum by a rotary evaporator (Equitron, Roteva, Germany).

  • In table 2, color values are not measured visually, but with a device, such as Hunter Lab.

AU Color, smell, and texture were evaluated sensory to choose the appropriate type for manufacturing. Yes, the color is measured physically by the Hunter device. Thank you for the clarification. The table has been modified from physical properties to sensory properties.

  • The purpose of the study is not fully understood. Extending the shelf life of these commercial sauces or increasing their biological activity? Why infertile was chosen as the biological activity test in the study?

AU The research was not designed for commercial purposes but to fortify sauces with propolis powder. As such, the assessment of biological activity is the main aim of the research. As for infertility, the purpose of this study was to test the efficiency of these products as a functional food to prevent toxicity caused in the testis because of exposure to cadmium. In the text, we stressed these concepts ("Infertility is a significant issue in contemporary societies that impacts many couples globally. There is growing interest in the role of environmentally hazardous metal exposures in reducing sperm concentration and male fertility in humans [3]. Male factors contribute to infertility in about 40–50% worldwide [4]. The biological effects of heavy hazardous metals such as cadmium (Cd) on the human body are well-known [5]. Over time, the accumulation of cadmium in different visceral organs such as the liver, lungs, kidney, and testis altered many physiological processes and the histological structure of the testicular tissue of various animal species. People are exposed to Cd through tobacco use, food, and industrial occupational and environmental pollution [5]. For this reason, the aim of this study was to produce a dipping sauce powder fortified with propolis as a functional food to reduce testicular toxicity induced by cadmium chloride (CdCl2) using technological, biochemical, histological, and morphometric studies"). Looking forward, the sauces evaluated were designed for use in human nutrition, whereby sensory evaluation and, most importantly, shelf life, were evaluated to test the potential of adding propolis on the usability of the sauces.

Round 2

Reviewer 1 Report

The manuscript has been improved, all the comments had been taken into account.

Author Response

The comments are in the attached file

Reviewer 3 Report

There are two missing points in this study, the sensory properties, because the taste of propolis deeply affects the flavor of the sauce, and the other is the insufficient biochemical parameters. The status of liver enzymes is unclear. Cadmium toxicity should have been demonstrated not only in the testicles but also in the liver.

Also, the analysis of propolis is a bit problematic.  Total  fat  was found 21% and 42%, these values are not actually oil, it is a waxy wax that dissolves in oil.

Also, why Chinese propolis has a lower phenolic composition, but has a higher DPP radical scavenging ability, there is a calculation error here. The SC50 value for DPPH needs to be defined, and how it is found should be explained.

After the properties of the propolis in the study are given more precisely, the total amount of phenolic substance in the prepared extract should be given per ml of the etbolic extract. You can get help from this article;

-European Food Research and Technology, 249(5), 1213-1233.

It was not presented as a cover letter made in the article. No explanation has been given. Only the main text has been edited. The article should be prepared in a more understandable way.

Author Response

The comments are in the attached file
